# Entanglement Negativity and Concurrence in Some Low-Dimensional Spin Systems

**DOI:** 10.3390/e24111629

**Published:** 2022-11-10

**Authors:** Leonardo S. Lima

**Affiliations:** Department of Physics, Federal Center for Technological Education of Minas Gerais, Belo Horizonte 30510-000, MG, Brazil; lslima@cefetmg.br

**Keywords:** entanglement negativity, triangular lattice, metal-insulating antiferromagnet, bicubic interaction, 75.10.Pq, 75.40.Mg

## Abstract

The influence of magnon bands on entanglement in the antiferromagnetic XXZ model on a triangular lattice, which models the bilayer structure consisting of an antiferromagnetic insulator and normal metal, is investigated. This effect was studied in ferromagnetic as well as antiferromagnetic triangular lattices. Quantum entanglement measures given by the entanglement negativity have been studied, where a magnon current is induced in the antiferromagnet due to interfacial exchange coupling between localized spins in the antiferromagnet and itinerant electrons in a normal metal. Moreover, quantum correlations in other frustrated models, namely the metal-insulation antiferromagnetic bilayer model and the Heisenberg model with biquadratic and bicubic interactions, are analyzed.

## 1. Introduction

Recently, topologic phase transitions (TPT) have been an important subject in condensed matter physics because this class of phase transition cannot be described by the theory of spontaneous symmetry breaking. Within this realism, the study of topological electronic systems has been extended to topological magnon insulators, which are elementary excitations above a ground state in a magnetically ordered system present in magnetic materials, where the band structure is expected to exhibit nontrivial topological properties [1,2,3,4,5,6,7,8,9,10,11,12,13,14,15,16,17]. Although magnons can exhibit the thermal Hall effect in contrast to electronic systems, there are no quantized Hall plateaus because magnons obey Bose–Einstein statistics. On the other hand, the study of quantum correlation, entanglement in quantum spin systems, as well as quantum Hall systems has been a unified field that has recently grown a lot. In this case, quantum information can be found by the study of the reduced density matrix, whose entanglement spectrum and von Neumann entropy are well recognized to contain key features. Furthermore, the solid-state proposals for quantum computation include quantum Hall systems (QHs), where one of the features of many QHs studied for quantum information processing is the scalability of their technology, which permits the creation of a large number of qubits [18]. The purpose of the present paper is to study quantum correlation using the entanglement negativity as a quantifier in important models of quantum magnetism and condensed matter physics [19].

*The half-integer and integer spin XXZ model with frustrating interactions:* It is well known that the XXZ model (anisotropic Heisenberg model) in lattice dimensions d=1,2 presents very different ground-state proprieties that depend on spin value; the geometry of the lattice, which includes different types of interactions with the lattice, namely the Dzyaloshinskii–Moriya interaction (DM); single-ion anisotropy; spin-phonon coupling, etc. For instance, it is well established that while the quantum integer spin one-dimensional XXZ model in the isotropic point presents a gap in the spectrum (Haldane’s gap) [20,21], the same model with a half-integer spin is gapless due to the Lieb, Schultz, and Mattis theorem [22]. Moreover, elementary excitations are different for an integer and half-integer spin. In the first case (integer spin), they are magnons, and in the second (half-integer spin), spinons. As pointed out in Ref. [23], the interactions between magnons or spinons with defects in the lattice and phonons have generated a number of intriguing features [24,25,26,27,28,29]. In addition, the spin–lattice interaction’s magnetic systems for example, leads to modifications in the spectrum of spin excitations [30,31,32] and to the formation of new phases, such as the spin-Peierls dimerized phase. Thus, we investigated the effects of different quantum and topological phase transitions induced by the spin-couplings, as well as the dimensionality and geometry of the lattice on quantum correlation and entanglement in Refs. [33,34,35,36,37,38,39,40,41,42,43,44,45,46]. The presence of different types of coupling leads to the different quantum and topological phases that are studied using analytical techniques, such as the nonlinear sigma model, spin-wave theory, and SU(N) Schwinger bosons, which are adequate in treating each case at low energy limits, as well as numerical techniques, such as the quantum Monte Carlo method and the density matrix renormalization group (DMRG), and so on. The quantum phase transition, which is induced by the Dzyaloshisnkii–Moriya interaction and external magnetic fields, and its effect on entanglement, has been investigated in one- and two-dimensional antiferromagnets in Refs. [34,35]. The effect of site dilution in the anisotropic two-dimensional XY model in the large anisotropy phase has been analyzed in Refs. [38,41]. Furthermore, the effect of quantum phase transition on quantum correlation in different spin-frustrated models, such as in the honeycomb and triangular lattices, has been analyzed in Refs. [36,40,43]. The effect of spin–phonon coupling on quantum correlation in the XY model is analyzed in Ref. [44]. The influence of magnon-band splitting and topological phase transitions, as well as the spin Nernst effect, has been investigated in Refs. [33,35,45,46]. The entanglement in a bilinear biquadratic model for iron-based superconductors has been studied in Refs. [34,35]. Here, we intend to study quantum correlation and entanglement in three different models: First, we study the quantum-frustrated spin model given by the triangular-lattice antiferromagnet with a external magnetic field, which presents four different quantum phases and magnon bands splitting that generates an effect in quantum entanglement. Next, we investigate quantum correlations in the fermion system given by the metal-insulating antiferromagnet bilayer model, in addition to quantum correlations in the two-dimensional Heisenberg model with biquadratic and bicubic interactions. We use different measures of quantum correlation and entanglement, which permits us to obtain the entanglement more easily in each case.

The paper is organized as follows: In Section 2, we discuss the XXZ model and its different phases on a triangular lattice.In Section 3, we present our numerical results for entanglement negativity as a function of *T* and anisotropy Δ. In Section 4, we analyze quantum entanglement in the metal-insulting antiferromagnetic bilayer model, which is an important model of spintronics. In Section 5, we study the effect of high-order terms used as the bicubic and biquadratic terms on quantum correlation in the Heisenberg model. In the last Section 6, we present our conclusions and final remarks.

## 2. XXZ Model on Triangular Lattice

The XXZ model on a triangular lattice with a magnetic field along the *z* axis is given by the spin Hamiltonian
(1)$$H=\sum_{\langle i\alpha ,j\beta \rangle }\left[JS_{i\alpha }\cdot S_{j\beta }+\Delta S_{i\alpha }^{z}S_{j\beta }^{z}\right]-\sum_{i\alpha }H_{\alpha }S_{i\alpha }^{z},$$
where $S_{i\alpha }$ is the spin operator localized at the αth sublattice of the ith unit cell in a triangular lattice and Δ>0 indicates an anisotropic exchange coupling. The notation α,β=A,B,C denotes the three sublattices in consideration, where 〈···〉 denotes the summation over the nearest-neighbor lattice sites. Moreover, the first term represents the magnetic exchange interaction with J>0 (antiferromagnet). The last term is an externally applied magnetic field.We consider the energy in *J* units (J=1) and ω=ℏ=c=1. The second term with Δ>0 indicates the anisotropy exchange, which is Δ=1 for the isotropic Heisenberg model. We consider that $H_{\alpha }$ depends on the sublattice. For 0<H<3, the system is in the *Y* phase, and for $3<H<H_{cl}$, the system is in the up-up-down phase (UUD). The phase boundary between the UUD and V phases is given by $H_{cl}=\frac{3}{2}\left[1+2\Delta +\sqrt{1+12\Delta +4\Delta^{2}}\right]$. For $H_{cl}<H<9+2\Delta$, the system is in the *V* phase and for H>9+2Δ, the system is in the fully polarized phase (FP). If we make the magnetic field on the *B* sublattice as $H_{B}=H+\varepsilon$ and ε≪1, the *Y* and *V* phases are replaced by distorted ones. The lattice considered with the different phases are represented in Figure 1.

*Linear spin-wave theory*: In the linear spin-wave approach (LSW), we perform a local rotation in the coordinate system at each sublattice at each lattice point, so that the mean-field directions of the spins point along the local *z* axis
(2)$$\begin{array}{c} S_{j\alpha }=\left(\begin{array}{ccc} \cos\theta_{\alpha } & 0 & \sin\theta_{\alpha }\\ 0 & 1 & 0\\ -\sin\theta_{\alpha } & 0 & \cos\theta_{\alpha } \end{array}\right){\tilde{S}}_{j\alpha }. \end{array}$$
In the harmonic approximation, we perform the Holstein–Primakoff transformation
(3)$$\begin{array}{c} \;{\tilde{S}}_{j\alpha }^{+}\approx \sqrt{2S}a_{j\alpha }^{†}\left(1-\frac{a_{j\alpha }^{†}a_{j\alpha }}{2S}\right),\;{\tilde{S}}_{j\alpha }^{-}\approx \sqrt{2S}a_{j\alpha },\;{\tilde{S}}_{j\alpha }^{z}=S-a_{j\alpha }^{†}a_{j\alpha }, \end{array}$$
where $a_{j\alpha }^{†}\left(a_{j\alpha }\right)$ are creation (annihilation) boson operators. The Hamiltonian is written in the momentum space as
(4)$$\begin{array}{c} H=E_{0}+\frac{S}{2}\sum_{k}\left(\begin{array}{cc} \phi^{†}\left(k\right) & \bar{\phi }\left(-k\right) \end{array}\right)H\left(k\right)\left(\begin{array}{c} \phi (k)\\ {\bar{\phi }}^{†}\left(-k\right) \end{array}\right), \end{array}$$
where $E_{0}$ is the energy of the ground state in the mean-field approach and $\phi^{†}\left(k\right)=\left(a_{k,A}^{†},a_{k,B}^{†},a_{k,C}^{†}\right)$, where the tilde over ϕ means to transpose. H(k) is given by
(5)$$\begin{array}{c} H\left(k\right)=\left(\begin{array}{cc} A(k) & B(k)\\ B^{*}\left(-k\right) & A^{*}\left(-k\right) \end{array}\right), \end{array}$$
where
(6)$$\begin{array}{c} \;{\left[A\left(k\right)\right]}_{\alpha \alpha }=H_{\alpha }\cos\theta_{\alpha }-\sum_{\alpha \neq \beta }\left[3\cos\left(\theta_{\alpha }-\theta_{\beta }\right)+3\Delta \cos\theta_{\alpha }\cos\theta_{\beta }\right],\\ \;{\left[A(k)\right]}_{\alpha \delta }=\left[\cos\left(\theta_{\alpha }-\theta_{\beta }\right)+\Delta \sin\theta_{\alpha }\sin\theta_{\beta }+1\right]\Gamma_{\alpha \beta }\left(k\right),\\ \;{\left[B(k)\right]}_{\alpha \beta }=\left[\cos\left(\theta_{\alpha }-\theta_{\beta }\right)+\Delta \sin\theta_{\alpha }\sin\theta_{\beta }-1\right]\Gamma_{\alpha \beta }\left(k\right)\\ \;{\left[B(k)\right]}_{\alpha \alpha }=0, \end{array}$$
where
(7)$$\Gamma_{\alpha \beta }\left(k\right)=2\cos\left(k_{x}/2\right)\cos\left(\sqrt{3}k_{y}/2\right).$$
Due to time-reversal symmetry, we have $A\left(k\right)=A^{*}\left(-k\right)$ and $B\left(k\right)=B^{*}\left(-k\right)$. The Hamiltonian above can be diagonalized by a paraunitary Bogoliubov transformation, producing a matrix $T_{k}$ such that $\omega_{k}=T_{k}^{†}H\left(k\right)T_{k}$, where $T_{k}^{†}H\left(k\right)T_{k}=\mathrm{diag}(\omega_{1,k},\omega_{2,k},\omega_{3,k},\omega_{1,-k},$$\omega_{2,-k},\omega_{3,-k}),$ The magnon energy bands are given by the diagonalization problem of the bosonic Hamiltonian Equation (Equation 4) [47]
(8)$$\begin{array}{c} \det\left(H\left(k\right)-\omega_{k}I\right)=0 \end{array}$$
or
(9)$$\begin{array}{c} \left|\begin{array}{cc} A\left(k\right)-\omega_{k}I & B(k)\\ B^{*}\left(-k\right) & A^{*}\left(-k\right)-\omega_{k}I \end{array}\right|=0, \end{array}$$
in which I is an *m*-square unit matrix. Hence, we obtain
(10)$$\begin{array}{c} \left(A\left(k\right)-\omega_{k}I\right)\left(A^{*}\left(-k\right)-\omega_{k}I\right)-B\left(k\right)B^{*}\left(-k\right)=0\\ \omega_{k}=\frac{1}{2}[\left(A\left(k\right)+A^{*}\left(-k\right)\right]\\ \pm \sqrt{{\left(A\left(k\right)+A^{*}\left(-k\right)\right)}^{2}-4\left(A\left(k\right)A^{*}\left(-k\right)-B\left(k\right)B^{*}\left(-k\right)\right)}]. \end{array}$$
In a uniform magnetic field, the system is in the *Y* phase and the lowest magnon bands touch each other at *K* and $K^{\prime }$ points, generating massless Dirac-cone-like dispersions. The mean-field ground state is at $\theta_{0,\alpha =A}=\pi$, $\theta_{0,\alpha =B}=\theta_{0,\alpha =C}=\theta$, where
(11)$$\theta =\cos^{-1}\left(\frac{3\Delta +h+3}{3\Delta +6}\right)$$
is determined by minimizing the mean-field energy given by
(12)$$\begin{array}{c} E_{MF}=3S^{2}\sum_{\langle \alpha ,\beta \rangle }\left[\left(1+\Delta \right)\cos\theta_{\alpha }\cos\theta_{\beta }+\sin\theta_{\alpha }\sin\theta_{\beta }\right]\\ -\sum_{\alpha }H_{\alpha }\cos\theta_{\alpha }, \end{array}$$
where the pair α,β means summation over the pairs (A,B), (B,C), and (C,A). $\theta_{\alpha }$ is determined by minimizing the mean-field energy $E_{MF}$, which leads to different solutions for $\theta_{\alpha }$, identifying the different phases nominated as *Y*, *V*, UUD, and FP [48]. For 0<H<3, the system is in the *Y* phase, which we are mainly concerned with. Moreover, we focused on the antiferromagnetic region Δ>0, where there are four different phases in a uniform magnetic field $H_{\alpha }=H$.

## 3. Quantum Correlation and Entanglement Negativity

Quantum entanglement is the quantum mechanical property that Schrödinger singled out many years ago as “the characteristic trait of quantum mechanics”; furthermore, it has been analyzed many times in connection to Bell’s inequality [49,50]. A pure pair of quantum systems is called entangled if it is unfactorable, and if conveyed, a mixed state is entangled if it can not be represented as a mixture of factorable pure states. It is a well known fact that quantum information theory can be used together with condensed matter physics to characterize quantum phase transitions (QPT) that are related to the ground-state energy of quantum many-particle systems. Thus, quantifying quantum correlations in these many-body systems enhances condensed matter physics and quantum information theory because they are a measure of quantum correlations or entanglements in a system given by von Neumann entropy [51].

The von Neumann entropy (VN) of a quantum state ρ is defined by the formula $S\equiv -\mathrm{Tr}\left(\rho \log_{2}\rho \right)$, where we define $0\log_{2}0\equiv 0$ for the Shannon entropy [52,53]. Thus, we can define the quantum version of the entropy by the relative entropy of μ to ν, which is defined by $S\left(\mu ||\nu \right)\equiv \mathrm{Tr}\left(\mu \log_{2}\mu \right)-\mathrm{Tr}\left(\nu \log_{2}\nu \right)$, where the quantum relative entropy is non-negative $S\left(\mu ||\nu \right)\geq 0$, with equality if and only if μ=ν. The VN entropy is a quantifier of the entanglement between two different partitions of a system nominated as *A* and *B*. The ground state ${|\Psi \rangle }_{AB}$ belongs to a Hilbert space composed of $H=H_{A}⊗H_{B}$. Thus, following Schmidt’s decomposition procedure, we can write ${|\Psi \rangle }_{AB}=\sum_{i=1}^{N}\kappa_{i}|\varphi_{i}\rangle_{A}⊗{|\phi_{i}\rangle }_{B}$, where $\alpha_{i}$ are the Schmidt’s coefficients, and $N\leq \min(\dim H_{A},H_{B})$. In the following, the whole system is considered as a binary system, with the block of N spinning as sub-system *A*, and the rest of the chain spinning as sub-system *B* [18,54,55]. Thus, the VN entropy between the two partitions is defined by $S_{A}=S_{B}\equiv -\sum_{i=1}^{N}\kappa_{i}^{2}\log_{2}\kappa_{i}^{2}$, where $S_{A}$ is the entropy of the subsystem *A* and $S_{A}=S\left(\rho_{A}\right)\equiv -{\mathrm{Tr}}_{A}\left(\rho_{A}\log_{2}\rho_{A}\right)$.

In general, the system will thermalize when the Gibbs distribution ρ is given by $\rho \propto e^{-\beta H}$, where the statistical ensemble describing the system for a long time is expected to be the canonical ensemble, being the density matrix of the canonical ensemble given by refs. [56,57,58,59].

$\rho =\frac{e^{-\sum_{{}_{k}}\omega_{k}{\hat{n}}_{k}}}{Z}$, where Z is the partition function. Thus, the density entropy s(k) that each mode contributes to thermodynamic entropy in the thermodynamic limit is given by $S/N^{2}\equiv \int_{-\pi }^{\pi }\int_{-\pi }^{\pi }s\left(k\right)d^{2}k$. Moreover, in the infinite time limit, the thermodynamic entropy and the VN entropy have the same density, representing the contribution that each mode has to the quantum entanglement. Consequently, in the large time limit, the results of the VN entropy must be equal to classical thermodynamic entropy since the results of spin-wave theory are accurate in this limit N→∞; therefore, the identification between the two kinds of entropies can be made [59].

*Entanglement negativity*: The entanglement negativity is the linear and partial transpose where the trace norm is a convex and monotone function; however, it is not additive. Furthermore, it presents a large deficiency, i.e., a failure in satisfying the discriminant property, with the entanglement E(ρ)=0 if and only if ρ is separable [60,61]. The entanglement negativity [19,61,62] is given for a mixed state $\rho_{GE}$ by
(13)$$\begin{array}{c} N\left(\rho \right)=\frac{∥\rho_{A}^{T}{∥}_{1}-1}{2}, \end{array}$$
where $\rho_{A}^{T}$ is the partial transpose of $\rho_{GE}$ with respect to subsystem *A*, and ${∥\cdot \cdot \cdot ∥}_{1}$ is the trace norm. The logarithmic negativity [63]
(14)$$E_{N}\left(\rho \right)=\log_{2}{∥\rho_{A}^{T}∥}_{1},$$
is used much more often as a measure of thermal entanglement for disjoint intervals. Consequently, the negativity has been proven to be useful in detecting topological orders [64,65], where one makes $\rho_{A}=\rho_{GE}$, which is the entanglement negativity given by $E_{N}\left(\rho \right)=-\log_{2}∥\frac{e^{-\beta \sum_{{}_{k}}\omega_{k}{\hat{n}}_{k}}}{Z}∥=\beta \sum_{k}\omega_{k}{\hat{n}}_{k}+\log_{2}\prod_{k}\left(1+e^{-\beta \omega_{k}}\right)$.

In Figure 2, we obtain the entanglement negativity as a function of *T*. We obtain a divergence of $E_{N}\left(\rho \right)$ at T→0 due to large increases in the quantum fluctuations near T=0; therefore, there is a loss of quantum information at this limit. The behavior at the high *T* range is only qualitative due to the limitations of the spin-wave approach we used. The inset of the figure shows the variation with Δ for two magnon bands: $\omega_{k}^{+}$ and $\omega_{k}^{-}$. In Figure 3, we display the behavior of the entanglement negativity as a function of Δ. We obtain that the entanglement negativity is finite at the XY limit (Δ=0), where the terms of scattering between the quasiparticles given by the term $S_{i}^{z}S_{j}^{z}$ are absent. We obtain an increase of $E_{N}\left(\rho \right)$ with Δ up to the isotropic limit Δ=1. The behavior of the quantum correlations is determined by the behavior of energy bands that depend on the coupling parameters of the model, which generates a large effect on entanglement negativity. The mean-field ground-state solution of $\theta_{\gamma }$ can be expanded as $\theta \approx \theta_{\gamma ,0}+\Delta_{\gamma }\varepsilon$, where the results of this expansion are very near to the exact numerical method [48]. Consequently, the results of $E_{N}\left(\rho \right)$ obtained with the spin-wave approach are near to the numerical results.

## 4. Metal-Insulting Antiferromagnet Bilayer Model

Antiferromagnetic insulators have generated interest as possible alternatives to ferromagnetic insulators as active components in spintronics [66,67,68]. A model of interest consists of a bilayer structure comprising an antiferromagnetic insulator on top of a normal metal. Voltage bias is applied to the normal metal in order to produce an electron current along the *x* axis. The electrons interact with the spins, leading to an induced magnon spin current [68]. We consider the system illustrated in Figure 4 to be two-dimensional and apply square lattice models. We start out from a tight-binding description of electrons hopping between lattice sites in a normal metal. For the antiferromagnetic insulator, we consider localized spins with easy-axis anisotropy, which interact with each other through a nearest-neighbor exchange interaction and a next-nearest-neighbor interaction. For a sufficiently small and isotropic Fermi surface in the normal metal, the Hamiltonian describing the electrons takes the form $H_{N}=\sum_{k\sigma }\varepsilon_{k\sigma }c_{k\sigma }^{†}c_{k\sigma }$, where $\varepsilon_{k\sigma }=tk^{2}a^{2}-\mu -\sigma H_{e}$ and $c_{k,\sigma }^{†}$ is the operator for an electron with momentum k and spin σ=↑,↓. *t* is the electron-hopping amplitude, *a* is the lattice constant, μ is the chemical potential, and $H_{e}$ is a spin-splitting field.

The Hamiltonian describing the magnons is given by
(15)$$\begin{array}{c} H=\sum_{k}\left(\xi_{k}+h_{\nu }\right)\alpha_{k}^{†}\alpha_{k}+\left(\xi_{k}-h_{\nu }\right)\beta_{k}^{†}\beta_{k}, \end{array}$$
where we consider the lattice spacing a=1. The magnon spectrum is given by $\xi_{k}=\sqrt{\Delta +\mu^{2}k^{2}}$, where Δ is the gap in the spectrum. $h_{\nu }$ is a splitting of the magnon modes through an external field. $\alpha_{k}^{†}$ is the creation operator for a spin-down and $\beta_{k}^{†}$ is the creation operator for a spin-up. The model is represented in Figure 4. In Figure 5, we display the behavior of the entanglement negativity as a function of the external field *h* and the gap of the magnon spectrum Δ, which generates a splitting of the magnon modes, where we consider $\Delta =h_{\nu }=h$. The aim is to verify the effect of splitting the magnon modes introduced through an external field on quantum correlation. As we can see, the quantum correlation increases with an increase in the magnon bands’ splitting. The same behavior occurs with increases in the chemical potential μ, where the magnon bands’ splitting becomes higher. In addition, we discover that the entanglement negativity tends toward a finite value for the model without a magnetic field and, hence, Δ=0.

## 5. Two-Dimensional Heisenberg Model with Bilinear–Biquadratic–Bicubic Terms

The interest in spin Heisenberg models with spin S>1/2 started many years ago, with valence bond states serving as a toy model related to high-$T_{c}$ superconductivity [69]. The AKLT model extended the notion of valence bond states to spins higher than 1/2. The AKLT model for $J_{2}/J_{1}=1/3$ has $J_{1}$ and $J_{2}$ as both positive, and the ground state is exactly solvable, which is defined as
(16)$$H_{AKLT}=\sum_{\langle i,j\rangle }S_{i}\cdot S_{j}+\frac{1}{3}\sum_{\langle i,j\rangle }{\left(S_{i}\cdot S_{j}\right)}^{2}.$$
The higher-order Heisenberg model for any spin-*S* can be written generically as
(17)$$H=\sum_{\langle i,j\rangle }\sum_{\nu =1}^{2S}J_{\nu }{\left(S_{i}\cdot S_{j}\right)}^{\nu }.$$
Thus, for the bilinear–biquadratic–bicubic model with S=3/2, the higher-order Heisenberg Hamiltonian reads:(18)$$H_{3/2}=\sum_{\langle i,j\rangle }\left[J_{1}S_{i}\cdot S_{j}+J_{2}{\left(S_{i}\cdot S_{j}\right)}^{2}+J_{3}{\left(S_{i}\cdot S_{j}\right)}^{3}\right].$$
For S=3/2, the diagonalization of this Hamiltonian for a two-site system gives four energy levels [70]
(19)$$\begin{array}{ccc} E_{1} & = & -\frac{15}{64}\left(16J_{1}-60J_{2}+225J_{3}\right),\;\mathrm{Sin}\mathrm{glet}\; \end{array}$$
(20)$$\begin{array}{ccc} E_{2} & = & -\frac{11}{64}\left(16J_{1}-44J_{2}+121J_{3}\right),\;\mathrm{Triplet}\; \end{array}$$
(21)$$\begin{array}{ccc} E_{3} & = & -\frac{3}{64}\left(16J_{1}-12J_{2}+9J_{3}\right),\;\mathrm{Quintuplet}\; \end{array}$$
(22)$$\begin{array}{ccc} E_{4} & = & \frac{9}{64}\left(16J_{1}+36J_{2}+81J_{3}\right),\;\mathrm{Sin}\mathrm{glet}\;. \end{array}$$
Considering an offset energy, as in the case of an integer spin, where $E_{\mu }^{\prime }=E_{\mu }+E_{off}$, μ=1,2,3,4, $E_{off}$ is an offset energy [70], and using Equation (Equation 19), the analytical expressions for rations of exchange constants $J_{2}/J_{1}$ and $J_{3}/J_{1}$ in terms of $E_{\mu }^{\prime }$ for S=3/2 are given in Ref. [70] for two and three orbitals per site using exact diagonalization. The aim here is to study the quantum correlation for the model (Equation (Equation 18)). We use a density matrix renormalization group (DMRG), which is a powerful numerical technique that is adequate for obtaining the ground-state energy for this model. For the model in Equation (Equation 18), including the biquadratic term, the von Neumann entropy was studied in Refs. [34,35]; however, the calculation of the von Neumann entropy or entanglement negativity for this model, including the bicubic term, is a more difficult task and can be performed in a future study.

### 5.1. Analysis by DMRG

DMRG is a well-known numerical technique suited to treating the one-dimensional spin-1/2 Heisenberg model [71,72]. However, any finite two-dimensional lattice can be mapped onto a one-dimensional lattice, where the sites of the lattice are numbered and, therefore, long-range interactions are introduced. Because in mean field theories the dynamics of the operators $\vec{S}\left(\vec{r}\right)$ are omitted, the variational principle assumes an expectation value of the operator $\langle \vec{S}\left(\vec{r}\right)\rangle$, where we neglect the fluctuations and drop higher-order terms.

### 5.2. Concurrence

In general, the bipartite entanglement can be investigated using different quantifiers as entanglement negativity, fidelity, concurrence, and von Neumann entropy. The concurrence is a quantity that can be expressed in terms of correlation functions of entanglement particles, where the examination of the bipartite entanglement in the considered model can be performed. The concurrence of the density matrix of a pair of qubits 1 and 2 $\rho_{12}$ is the density matrix of either pure or mixed states and is defined by [73,74]
(23)$$C=\max\{0,\lambda_{1}-\lambda_{2}-\lambda_{3}-\lambda_{4}\},$$
in which $\lambda_{1},...,\lambda_{4}$ are the square roots of the eigenvalues, in decreasing order, of the operator $R_{1,2}=\rho_{12}\left(\sigma_{y}⊗\sigma_{y}\right)\rho_{1,2}^{*}\left(\sigma_{y}⊗\sigma_{y}\right)$, and $\lambda_{1}$ is the greatest square root of the four eigenvalues of $R_{1,2}$. This formula uses the “spin-flip” transformation, which is applicable to the states of an arbitrary number of qubits [73,74]. For a state of two qubits $\rho_{12}$, a nonzero concurrence means that qubits 1 and 2 are correlated or entangled. The concurrence C=0 corresponds to an unentangled state and C=1 corresponds to a maximally entangled state. For N qubits, we apply the above transformation to each individual qubit.

The relation between the concurrence and internal energy *U* of the system is given by C=1/2max[0,−U/(JN)−1] for the antiferromagnetic system (AFM) and C=1/2max[0,U/(3JN)−1] for the ferromagnetic system (FM) [58]. Thus, the entanglement is uniquely determined by the partition function of the system $|\langle \sigma_{z}⊗\sigma_{z}\rangle \left|=Z^{-1}\right|\sum_{i}e^{-\beta E_{i}}$$\langle i|\sigma_{z}⊗\sigma_{z}\left|i\rangle \right|\leq Z^{-1}|\sum_{i}e^{-\beta E_{i}}∥\sigma_{z}⊗\sigma_{z}|i\rangle ∥\leq Z^{-1}\sum_{i}e^{-\beta E_{i}}=1$, where we have −1≤U/(3JN)≤1. For the AFM case, the increase in the temperature or internal energy will generate a decreasing in the concurrence, and for a value such as −NJ, the concurrence will become zero. The temperature where the concurrence vanishes is called the threshold temperature.

In Figure 6, we present the concurrence *C* as a function of $J_{3}$ ($J_{3}<0$) for different values of $J_{2}$ ($J_{2}<0$) coupling using DMRG. The sign of biquadratic and bicubic strength ($J_{2}$ and $J_{3}$) depends on the relation between the energies $E_{1}$, $E_{2}$, $E_{3}$, and $E_{4}$. We have
(24)$$\begin{array}{cc} \frac{J_{2}}{J_{1}} & =\frac{4}{3}\frac{29E_{4}-85E_{3}+81E_{2}-25E_{1}}{\left(81E_{4}+115E_{3}-351E_{2}+155E_{1}\right)} \end{array}$$
(25)$$\begin{array}{cc} \frac{J_{3}}{J_{1}} & =\frac{4}{3}\frac{E_{4}-5E_{3}+9E_{2}-5E_{1}}{\left(81E_{4}+115E_{3}-351E_{2}+155E_{1}\right)}, \end{array}$$
The calculations were performed for a lattice size of N=192. We obtain a very small variation in the results with different lattice sizes N=64,128,192. Furthermore, we obtain a strong influence in the strength of the bicubic term $J_{3}$ on concurrence with the extinction of *C* near to $J_{3}\approx 1.1$. Thus, the analysis by DMRG in Figure 6 seems to confirm decreases in the concurrence of *C* with the coupling $J_{3}$.

## 6. Summary

In brief, we analyzed quantum correlation and entanglement in some low-dimensional quantum spin models, which are very important in condensed matter physics theory. We analyzed the triangular-lattice XXZ model with three sublattices denoted by *A*, *B*, and *C*, where we focused on the antiferromagnetic region Δ>0 and four different phases present in a uniform magnetic field $H_{\gamma }=H$. We focused on the *Y* phase for 0<H<3. The ground-state phase diagram of this spin model is obtained in the classical limit S→∞ with four different regions whose spin configurations are represented in Figure 2. Our results display a divergence of quantum correlation T=0 and a further strong effect of the anisotropy Δ on entanglement, generating an increasing of $E_{N}\left(\rho \right)$ from the XY limit up to isotropic limit. We analyzed the entanglement in a bilayer structure of an antiferromagnetic insulator on top of a normal metal—an important model for spintronic devices. Furthermore, we analyzed entanglement in the two-dimensional bicubic Heisenberg model, which is an important generalization of the AKLT model with higher-order corrections given by the bicubic term. For all models analyzed, we obtained strong entanglement variations with the coupling parameters. In a general way, in quantum spin systems, either real fields or complex fields generate a splitting of degenerate ground states, where the spins are aligned along the direction of the external magnetic field. However, the eigenvalues and eigenvectors of the real-spectrum system will not suffer many changes, with the external magnetic field and the initial state displaying oscillating behaviors periodically among all possible spin configurations.

## Figures and Tables

**Figure 1 entropy-24-01629-f001:**
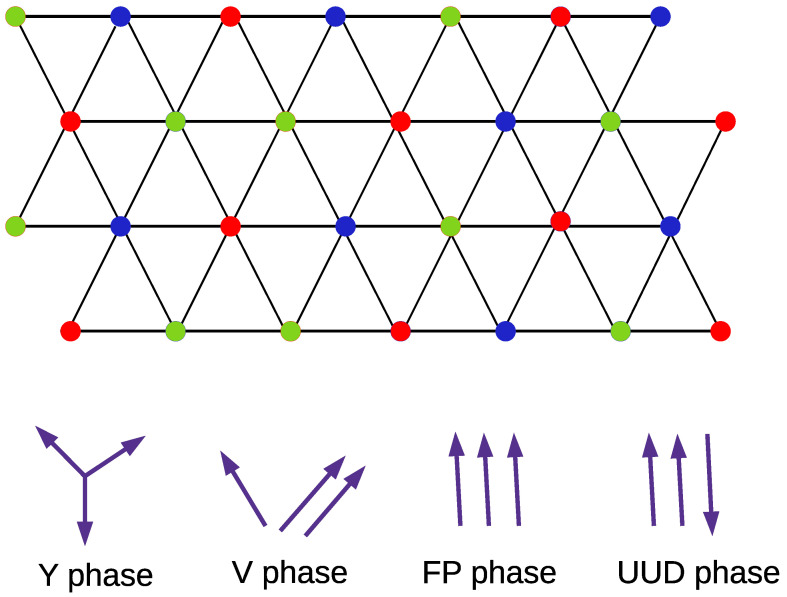
Representation of the triangular lattice for the model from Equation (Equation 1), with the three sublattices denoted as *A*, *B*, and *C*. Below, we have the four different spin orientations on *A*, *B*, and *C* sublattices.

**Figure 2 entropy-24-01629-f002:**
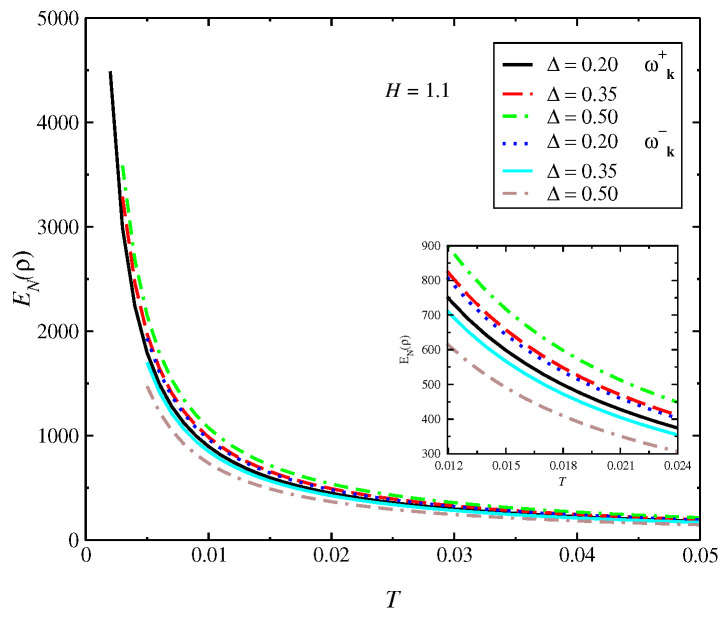
Entanglement negativity $E_{N}\left(\rho \right)$ vs. *T* for different values of Δ at H=1.1 for the model (Equation (Equation 1)). We obtain a small negative difference of the entanglement for the bands $\omega_{k}^{+}$ and $\omega_{k}^{-}$ (solution of Equation (Equation 9)). Furthermore, the entanglement negatively diverges at limit T→0 due to large increases in the quantum fluctuations near T=0, where a quantum phase transition takes place.

**Figure 3 entropy-24-01629-f003:**
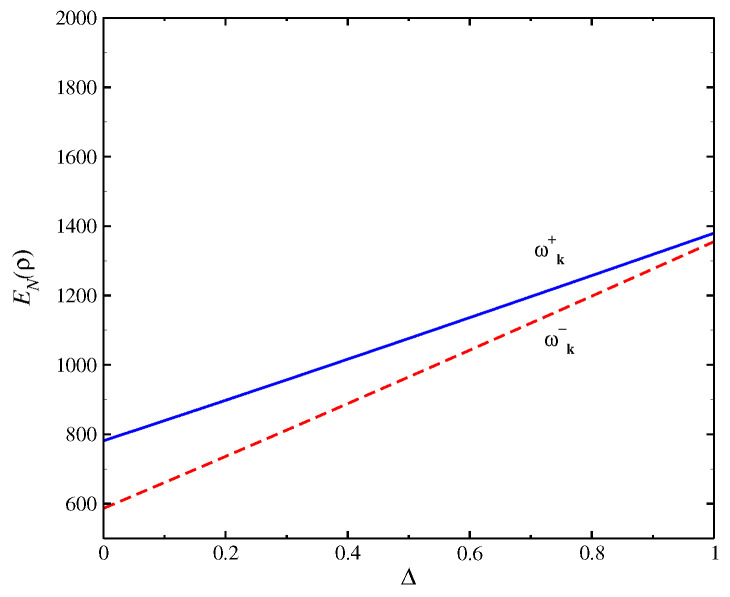
$E_{N}\left(\rho \right)$ as a function of Δ for H=1.1 for the model (Equation (Equation 1)). We find the negativity entanglement is finite at XY limit (Δ=0), increasing with Δ up to isotropic limit Δ=1.

**Figure 4 entropy-24-01629-f004:**
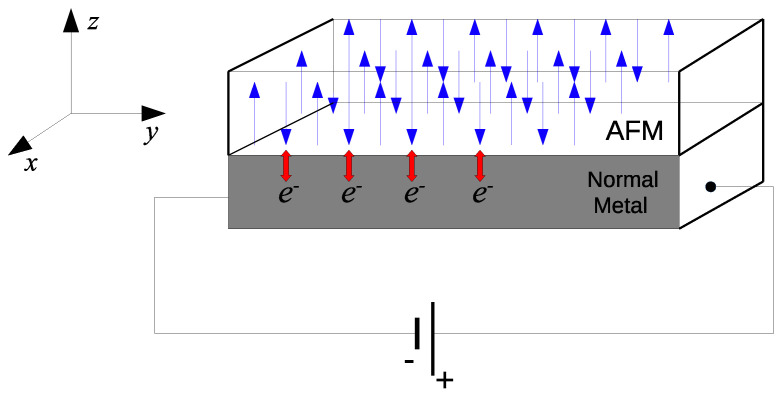
Bilayer structure consisting of an antiferromagnetic insulator on top of a normal metal of the model (Equation (Equation 15)). A voltage is applied to the normal metal in order to produce an electron current along the *x* axis.

**Figure 5 entropy-24-01629-f005:**
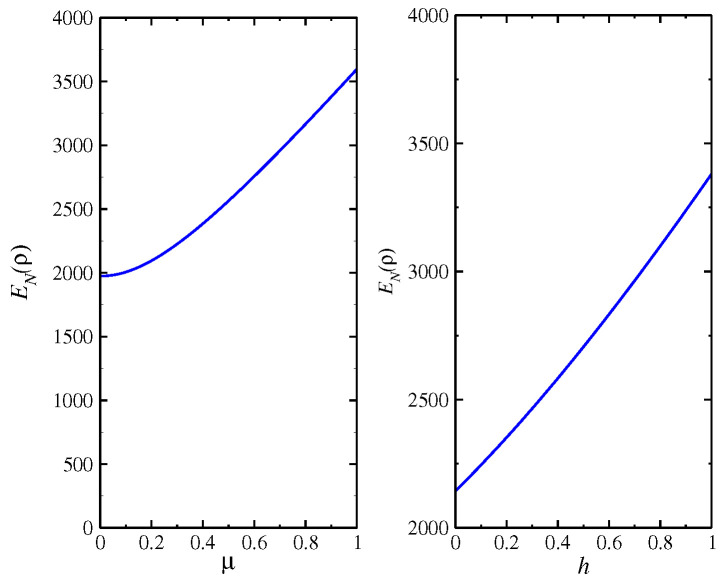
$E_{N}\left(\rho \right)$ for the model (Equation (Equation 15)), as a function of *h* (right-side) for a value of coupling μ as μ=0.9, and for a value of h=1.0 held fixed (left-side).

**Figure 6 entropy-24-01629-f006:**
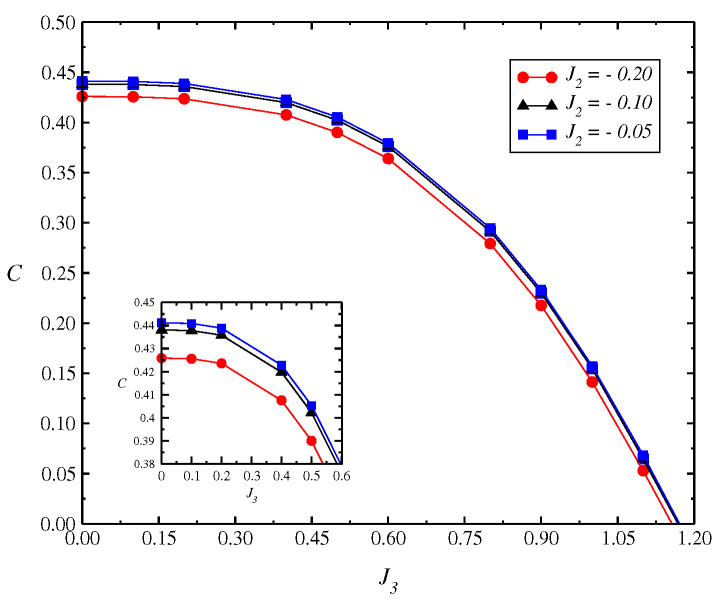
Concurrence C(ρ) vs. $J_{3}$ ($J_{3}<0$) for the model (Equation (Equation 18)) as a function of $J_{3}$ obtained by DMRG. We obtain a small change in the behavior for different values of $J_{2}$. The calculations were performed for a lattice size N=192 sites.

## Data Availability

All data generated or analyzed during this study are included in this paper.

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
