# Peer review of "Entanglement Negativity and Concurrence in Some Low-Dimensional Spin Systems"

_entropy, 2022, doi:10.3390/e24111629_

Round 1

Reviewer 1 Report

The author studied entanglement negativity and concurrence for a quantum system of ferromagnetic and antiferromagnetic triangular lattices.

This work is publishable in entropy, if the author revises it according to the following revision list.

1. In the introduction part, I recommend the following revision:

[old] ... the Refs.[20-22]. The paper is organized ...

[New] ... the Refs.[20-22].

The paper is organized ...

2. A revision may be necessary in Eq. (9).

3. For the phrase “the quantum relative entropy is non-negative S < 0“, in sec 3, the meaning of the English representation and equation are contradict each other.

4. In the equation after the sentence “Thus, from the Schmidt's decomposition procedure, we can write”, does i run 1 to N? If so, it may be revised as:

i -> i=1

5. The author wrote “negativity entanglement” in some parts, while he/she wrote “entanglement negativity” in other parts.

6. Before Eq. (15)

electromn -> electron

aplitude -> amplitude

7. Eq. (23) is always zero, if we consider the restriction imposed in lambda_i.

8. The author wrote “U/3JN”. Is it mean “U/(3JN)”? If so, a parenthesis may be necessary. Otherwise, I recommend to write it as “UJN/3”. 

Author Response

Dear reviewer, I thank you a lot by yours positive comments about the manuscript. 

I revised the manuscript  according to the  revision list. 

1) I revised  in the introduction part:

[old] ... the Refs.[20-22]. The paper is organized ...

[New] ... the Refs.[20-22].

The paper is organized ...

2) We corrected the Eq.(9) in the first and second lines.

3) I corrected the sentence  “the quantum relative entropy is non-negative S < 0“, in sec 3, to “the quantum relative entropy is non-negative S >= 0“ in this new version.

4) I corrected the equation after the sentence “Thus, from the Schmidt's decomposition procedure, we can write”. Yes, i runs 1 to N. I corrected the two equations after this sentence. 

5) We corrected  “negativity entanglement” in some parts of the manuscript to "entanglement negativity".

6) We corrected the typos pointed  before Eq. (15)

electromn -> electron

aplitude -> amplitude

7) We corrected the Eq. (23) and the line under it.

8) We corrected "U/3JN” to "U/(3JN)" as pointed.

9) We included more References in the Introduction section with an extra subsection explaining about the works made in other quantum spin systems. We read all the manuscript and corrected all typos that I found.

yours sincerely.

Reviewer 2 Report

I have struggled to read and understand this manuscript for various reasons. In all sections, from the introduction to the conclusions,  the author exposes collections of statements and even lines of mathematics without providing appropriate justifications or convincing passages, as if they were already done in other papers. The same description of the Hamiltonian model 1 is lacking when it refers to the notion of three sublattices without feeling the need to  make clear its literal meaning and physical context. Not to mention the absence of physical scenarios.  But there is another aspect that forced me to carefully read the bibliography of the manuscript.  In the introduction, at a certain point, the author, with reference to the purpose of the paper, writes:   The effect of topological phase transition on quantum correlation has been analyzed in other quantum spin systems in the Refs. [20-22].     The quoted references  are three papers by the same author published in 2021 in three different journals. Naturally I looked for them, being struck by the close proximity of the theme, of the models, of the mathematical tools and also of the problems studied even if they referred to a little bit different Hamiltonian models, all belonging, however, to a single family.
Intrigued by this circumstance, I looked at the bibliography of these three works, where I found other recent papers by the same author, sometimes co-authoring, once again exhibiting the same characteristics.
It seems to me that the author has identified an area in which it is easy to generate a class of similar problems referring to Hamiltonian models that can be said to belong to the same family.
Given that, I must say that I find surprising the succinct, almost invisible way in which the author has quoted this production of his.
I believe it is scientifically important and strictly necessary for a positive evaluation of the manuscript, a section entirely dedicated to what he has already published in the context of the manuscript  now submitted, highlighting that it is not a mere continuation or a marginal variation of things already done by the same author or  others.
In this regard, in section V, where Hamiltonian models of high spin are examined, I believe that the author must offer the reader a breath of Physics to make him participate in what is reported with too few details unfortunately.  Are these  results original? If so, this fact should be emphasized; if not, appropriate references should be introduced.  As for the references, I point out that I could not find reference 23, even going to look for it on PRB directly. Overall,  I think that the manuscript is linguistically poorly written, but above all the presentation is poorly written for the reasons mentioned above.  I therefore believe that the author should deeply modify the manuscript demonstrating the actual advancement made by his results in the field  under scrutiny.  But also the elimination of any doubt for each of the problematic aspects identified in this report is a necessary condition for the manuscript to be published on Entropy. In conclusion, the manuscript cannot be accepted without passing each of the points raised.

Author Response

Dear reviewer, thanks by yours comments.

1) I searched to improve the presentation of the manuscript as suggested.

2) I gave more explanation about the Hamiltonian Eq.(1) in the paragraph after the Hamitonian Eq.(1).

3) In the sentence "The effect of topological phase transition on quantum correlation has been analyzed in other quantum spin systems in the Refs. [20-22]."

I would want to say that I have analyzed the effect of quantum (QPT) and topological  phase transition (TPT) also in these References cited. Note that, each spin model is not only a "a little bit different Hamiltonian models, all belonging, however, to a single family." Although each Hamiltonian model seems matematically similar, each one represents to a family of different componds, containing a very different physics, with different quantum and topological phase transitions that suffers a large variation from a model to another. Moreover, each model is studied using  different analytical and numerical mathematical techniques. For instance, the one-dimensional Heisenberg model is very different from two-dimensional Heisenberg model where in the formel  the magnetic excitations are magnons while the second one, the magnetic excitations are spinons .  Furhtermore, the first was studied using the field theory of non linear sigma model where was found by F. D. M. Haldane that the spectrum presents a gap while for the same one-dimensional model given by the same Hamiltonian, of half-integer spin, although matematicaly similar was proved that is gapless in the spectrum, according of Lieb-Shultz and Mattis theorem. In addition, the  inclusion of different types of couplings as spin-phonon coupling, interaction between spins with the lattice as the Dzyaloshinskii-Moriya interaction and single-ion anisotropy that present a sizeable value in some componds as well as, considering different lattice geometries  may generate a large variety of  quantum and topological phase transitions each one canning be studied with a specific mathematical technicque. Hence, it is important to analyze the influence of the different types of couplings and lattices and hence QPT and TPT on quantum correlation and this was made in the References cited. Consequently, to explain better, I have added a subsection into the introduction section entirely dedicated to what I have already published in the context of the manuscript  now submitted, highlighting that this is not a mere continuation or a marginal variation of things already done by me or  others, where  I included the list of works published in the last years as well as an brief explaination about the results obtained in each one.

4)  We gave after Eq.(22) a more explain about the results, as suggested, with aim to detail more about the properties of the model that already was studied.

5) We corrected the Reference (23).

Yours sincerely

Round 2

Reviewer 1 Report

The author revised the manuscript according tp my comment.

I recommend the publication of this manuscript in Entropy.